# Iron deficiency and common neurodevelopmental disorders—A scoping review

Scout McWilliams[1], Ishmeet Singh[1], Wayne Leung[1], Sylvia Stockler[1,2,3]*, Osman S. Ipsiroglu[1,2,4]

1 H-Behaviours Research Lab (previously Sleep/Wake-Behaviour Research Lab), BC Children's Hospital Research Institute, Vancouver, Canada, 2 Department of Pediatrics, University of British Columbia, Vancouver, Canada, 3 Division of Biochemical Diseases, Department of Pediatrics, BC Children's Hospital, University of British Columbia, Vancouver, Canada, 4 Divisions of Child & Adolescent Psychiatry, Developmental Pediatrics and Respirology, Department of Pediatrics, Sleep/Wake-Behaviour Clinic at Sleep Program, BC Children's Hospital, University of British Columbia, Vancouver, Canada

☯ These authors contributed equally to this work.
* sstockler@cw.bc.ca

**Data Availability Statement:** All relevant data are within the paper and its Supporting information files.

## Abstract

### Background

A wealth of human and experimental studies document a causal and aggravating role of iron deficiency in neurodevelopmental disorders. While pre-, peri-, and early postnatal iron deficiency sets the stage for the risk of developing neurodevelopmental disorders, iron deficiency acquired at later ages aggravates pre-existing neurodevelopmental disorders. Yet, the association of iron deficiency and neurodevelopmental disorders in childhood and adolescence has not yet been explored comprehensively.

In this scoping review, we investigate 1) the association of iron deficiency in children and adolescents with the most frequent neurodevelopmental disorders, ADHD, ASD, and FASD, and 2) whether iron supplementation improves outcomes in these disorders.

### Method

Scoping review of studies published between 1994 and 2021 using "iron deficiency / iron deficiency anemia" AND "ADHD" OR "autism" OR "FASD" in four biomedical databases. The main inclusion criterion was that articles needed to have quantitative determination of iron status at any postnatal age with primary iron markers such as serum ferritin being reported in association with ADHD, ASD, or FASD.

### Results

For ADHD, 22/30 studies and 4/4 systematic reviews showed an association of ADHD occurrence or severity with iron deficiency; 6/6 treatment studies including 2 randomized controlled trials demonstrated positive effects of iron supplementation. For ASD, 3/6 studies showed an association with iron deficiency, while 3/6 and 1/1 systematic literature review did not; 4 studies showed a variety of prevalence rates of iron deficiency in ASD populations;

**Funding:** SM was supported by a summer studentship from BC Children's Hospital Research Institute (https://www.bcchr.ca/), IS was supported by funds from the BC Children's Hospital Research Institute for sleep wake behaviour research (to OI) and for treatable intellectual disability research (to SS). WL was supported by the Mach-Gaensslen Foundation (https://mach-gaensslen.ca/). The Iron Conundrum Workshop was funded by an award from the Michael Smith Foundation for Health Research (https://www.msfhr.org/). The funders had no role in study design, data collection and analysis, decision to publish, or preparation of the manuscript.

**Competing interests:** The authors have declared that no competing interests exist.

**Abbreviations:** AbBC, Aberrant behaviour checklist; ADHD, Attention deficit hyperactivity disorder; ADI-R, Autism Diagnostic Interview–Revised; ADOS, Autism Diagnostic Observation Schedule; ASD, Autism spectrum disorder; AuBC, Autism behaviour checklist; BI, Brain iron; BMI, Body mass index; CARS, Childhood autism rating scale; CBL, Child behaviour checklist; CGI-S, Clinical global impression severity; CHCM, Children's interview for psychiatric symptoms: parent version; CPRS, Conners parent rating scale; CRP, C-reactive protein; CRS, Conners rating scale; CSI-4, child symptoms inventory-4; CTRS, Conners teacher rating scale; ESR, Erythrocyte sedimentation rate; FASD, Fetal alcohol spectrum disorder; FEP, Free erythrocyte protoporphyrin; GDD, Global developmental delay; Hb, Hemoglobin; Hct, Hematocrit; ID, Iron deficiency; IDA, Iron deficiency anemia; IQ, Intelligence quotient; K-SADS, Kiddie schedule for affective disorders and schizophrenia; MCH, Mean corpuscular hemoglobin; MCHC, Mean corpuscular hemoglobin concentration; MCV, Mean corpuscular volume; MRI, Magnetic resonance imaging; NDD, Neurodevelopmental disorder; P-ChIPS, Cell hemoglobin concentration mean; RCT, Randomized controlled trial; RDW, Red cell distribution width; RLS, Restless legs syndrome; SCID-1, Structured clinical interview for DSM-IV-I; SDQ, Strengths and difficulties questionnaire; SF, Serum ferritin; SFI, sTfR/log ferritin index; SNAP, Swanson, Nolan, Pelham scale; sTfR, Soluble transferrin receptor; SWTD, Sleep wake transition disorders; T-DSM-IV-S, Turgay DSM-IV-Based Child and Adolescent Behaviour Disorders Screening and Rating Scale; TIBC, Total iron binding capacity; TSAT, Transferrin saturation; WURS, Wender Utah Rating Scale.

1 randomized controlled trial found no positive effect of iron supplementation on behavioural symptoms of ASD. For FASD, 2/2 studies showed an association of iron deficiency with growth retardation in infants and children with prenatal alcohol exposure.

## Conclusion

Evidence in favor of screening for iron deficiency and using iron supplementation for pediatric neurodevelopmental disorders comes primarily from ADHD studies and needs to be further investigated for ASD and FASD. Further analysis of study methodologies employed and populations investigated is needed to compare studies against each other and further substantiate the evidence created.

## Introduction

Neurodevelopmental disorders (NDDs) are a leading cause of morbidity in children, characterized by impairments in cognitive, adaptive and social functioning, and resulting in large burdens to patients, caregivers, and society [1]. Attention deficit hyperactivity disorder (ADHD), autism spectrum disorder (ASD), and fetal alcohol spectrum disorder (FASD) are currently the most frequently diagnosed NDDs. Notably, ADHD and ASD are increasingly seen in high-income countries such as Canada and the United States [2].

ADHD is defined as "impairing levels of inattention, disorganization, and/or hyperactivity-impulsivity" and is prevalent in up to 7% of children and adolescents [3]. ASD is characterized by persistent deficits in social and emotional interaction across multiple contexts, hampering normal development in young children and frequently progressing to mental health conditions in adulthood. The prevalence of ASD is 2.5% among US children [4]. FASD is a spectrum of conditions related to prenatal alcohol exposure, which results in physical, behavioural, and cognitive impairments. The global prevalence of FASD among children and youth in the general population is estimated to be 0.8%, but ranges significantly between countries and certain subpopulations with peak prevalence rates between 5 to 11% [5].

Iron deficiency (ID) has been positively associated with these NDDs with the common underlying pathophysiology being derived from iron's central role in the brain as a cofactor in neurotransmitter (dopamine and serotonin) synthesis, as well as in ATP production and myelination [6]. The impact of ID on the development and severity of NDDs depends on the developmental stage during which the brain is exposed to iron depletion. Intrauterine / maternal and early postnatal ID is likely to impact the development of brain structures, resulting in irreversible neurocognitive and behavioural deficits as the end point of the pathophysiological cascade [7]. The hippocampus appears particularly vulnerable, resulting in persistent hippocampus-based cognitive deficits in adulthood despite iron supplementation [8, 9]. Moreover, a recent study found that iron deficiency anemia (IDA) diagnosed earlier in pregnancy was associated with an increased risk of the development of ASD, ADHD, and particularly ID in offspring [10].

There is also evidence that postnatal ID may aggravate pre-existing NDDs. For ADHD, studies have shown that ID plays a role in the occurrence and the severity of hyperactive day and nighttime symptomology [11–14], and that iron supplementation is beneficial in the treatment of affected children [15]. Children with ASD are at particular risk for ID due to co-occurring selective food intake, food sensitivities, and gastrointestinal problems. Indeed, studies in children with ASD suggest a high prevalence of ID [16, 17] and a positive effect of iron

supplementation on sleep, one of the most debilitating morbidities in patients on the spectrum [18]. For FASD, experimental studies suggest a role of brain specific ID in FASD modulated by iron status of the dam [19], and observational studies support the similarities of behavioural phenotypes exhibiting challenging day and nighttime behaviours, including sleep [20].

According to the World Health Organization, ID is the most common and widespread micronutrient deficiency worldwide, affecting more than 30% of the world's population [21, 22]. However, despite all of the evidence outlined above, ID is not routinely investigated in patients with NDDs, including ADHD, ASD, and FASD, and iron supplementation has not been considered as a potentially effective treatment strategy.

We performed a scoping review with a twofold aim: first to assess existing evidence of an etiologic relationship between postnatal ID and prevalence and severity of ADHD, ASD, and FASD; second to determine effects of iron supplementation on outcomes of these NDDs. Furthermore, we were interested in the types of iron markers used for the identification of ID.

## Methods

### Literature search

We performed a literature search in the electronic databases MEDLINE (OVID), CINAHL (EBSCOhost), PsycINFO (EBSCOhost), and EMBASE (OVID) from 1994 to 2017, using the search terms "iron deficiency an(a)emia" AND "ADHD" OR "autism" OR "FASD" (Table 1), on February 3rd, 2017.

We updated our review in July 2019 by performing a literature search in the electronic databases MEDLINE (OVID), CINAHL (EBSCOhost), PsycINFO (EBSCOhost), and EMBASE (OVID) from January 1st 2016 to July 20th, 2019 using the same search terms "iron deficiency anemia" AND "ADHD" OR "autism" OR "FASD". In order to optimize the yield for treatment studies, we amended these original search terms by "iron supplementation" OR "iron substitution" OR iron replacement" OR "iron fortification" OR "micronutrient", but no additional references were identified. We repeated the entire search in August, 2020, and again in May, 2021 to provide the most updated information by the time of acceptance for publication. Here we have combined the results of all these searches.

**Table 1. Detailed search strategies used in each database.**

| | Medline (OVID) | Embase (OVID) | CINAHL (EBSCOhost) | PsycINFO (EBSCOhost) |
|---|---|---|---|---|
| Iron | **Anemia, Iron deficiency/** OR iron metabolism disorders/ OR iron/bl [blood] OR iron/df [deficiency] OR exp ferritins/bl, df [blood, deficiency] | **exp iron deficiency anemia/** OR (iron adj2 deficiency) OR (iron adj2 deficient) | **MH "Anemia, Iron Deficiency"** OR MH "ferritin/BL/DF" OR MH 'Iron/DF" OR anemia OR anaemia OR iron deficiency | **DE "iron"** OR **DE "anemia"** OR iron status OR ferritin OR ((anemia OR anaemia) AND iron deficiency) |
| ADHD | **exp "Attention deficit and disruptive behaviour disorders"/** attention deficit OR ADHD OR | **attention deficit disorder/** OR (attention deficit or ADHD) | **MH "attention deficit hyperactivity disorder"** OR adhd OR attention deficit | **DE "Attention deficit disorder with hyperactivity"** OR attention deficit OR ADHD |
| ASD | **exp Child Development Disorders, Pervasive/** OR autism OR autistic OR ASD | **exp autism/** OR (autism or autistic or ASD) | **MH "autistic disorder"** OR autism OR autistic OR asd OR autism spectrum disorder OR | **DE "Autism spectrum disorders"** OR DE autism OR autistic OR ASD |
| FASD | **Prenatal exposure, delayed effects/** OR **fetal alcohol spectrum disorders/** OR prenatal alcohol exposure OR PAE OR fetal alcohol syndrome OR foetal alcohol syndrome OR fetal alcohol spectrum OR foetal alcohol spectrum OR FASD | **fetal alcohol syndrome/** OR prenatal alcohol exposure OR PAE OR fetal alcohol syndrome OR foetal alcohol syndrome OR fetal alcohol spectrum OR foetal alcohol spectrum OR FASD | **MH "Fetal alcohol syndrome"** OR ((fetal OR foetal) AND alcohol) OR prenatal alcohol exposure OR PAE OR FASD | **DE "fetal alcohol syndrome"** OR prenatal alcohol exposure OR PAE OR fetal alcohol syndrome OR foetal alcohol syndrome OR fetal alcohol spectrum OR foetal alcohol spectrum OR FASD |

Bolded words are subject headings.

## Inclusion and exclusion criteria

The inclusion criteria for this review were 1) the study discusses the iron status of patients in association with their pre-existing diagnosis of ADHD, ASD, FASD; 2) the iron status is determined quantitatively using one or more iron markers that are primarily reflecting iron metabolism (e.g. serum ferritin (SF), transferrin saturation (TSAT), total iron binding capacity (TIBC) (primary iron markers)); 3) study methodologies including case reports, case series, case controls, cross-sectional studies, cohort studies, clinical trials, controlled clinical trials, randomized controlled trials (RCT), and systematic reviews.

References were excluded if: 1) subjects did not have iron deficient status reported; 2) iron status was not reported quantitatively using primary iron markers; 3) iron status of subjects was determined using only red blood cell (RBC)-related markers such as hemoglobin (Hb), mean corpuscular volume (MCV), or measurements of serum iron (SI) only; 4) the NDD was reported in association with prenatal / maternal / intrauterine ID; 5) study was not published in English; 6) full-length text was not available; 7) specific study types, including non-systematic reviews, dissertations, animal studies, correspondences, editorials, conference proceedings, health letters.

## Study selection

Three investigators independently selected the studies (SM, IS, WL) and 3 investigators (SM, IS, WL) classified the studies. SS and OI were consulted to settle any disagreement on study selection.

## Classification of study types

Studies identified after application of inclusion and exclusion criteria were subdivided:

1. According to the NDD investigated: ADHD, ASD, FASD.

2. Studies within each NDD were subdivided into:

   a. Association studies describing an association between ID and the respective NDD. Studies which described the frequency of ID within an NDD population, not supported by further statistical analysis, were termed as "prevalence studies".

   b. Treatment studies describing the effects of iron supplementation on symptoms of the respective NDD.

   c. Systematic reviews.

*Association studies* were classified as case control, case series, cohort- and cross sectional studies [23]. A positive association was considered if a study showed a relationship between one of the 3 NDDs and the presence of ID indicated by at least one primary iron deficiency marker. *No association* was considered if the studies did not show a relationship between ID and one of the 3 NDDs. Association studies were rated as positive associations when an association between ID and an NDD was supported through statistical analysis such as 1) between subjects with and without the neurodevelopmental disorder, 2) between subjects with and without ID, 3) correlation between iron levels and symptom severity, 4) odds ratio showing increased odds of having a NDD with ID. Studies reporting rates of ID in individuals with an NDD without further correlation studies were separated as prevalence studies.

*Treatment studies* were categorized *as experimental studies*, which included clinical trials, controlled clinical trials, and RCTs; and *non- experimental studies*, which included observational case series, and case reports. Experimental studies were analyzed for methodological

quality using the Jadad scale [24], in which one point was awarded for each of 1) through 3). One additional point was awarded if the study fulfilled each of 4) or 5), while and one point was deduced for 6) and 7), yielding a maximum score of five.

1. The study was described as randomized (this includes the use of words such as randomly, random, and randomization).

2. The method used to generate the sequence of randomization was described and appropriate (e.g. use of computer program to generate random allocations).

3. The study was described as double blinded.

4. The method of double blinding was described and appropriate (e.g. use of identical placebo).

5. The study included a description of withdrawals and dropouts.

6. The method used to generate the sequence of randomization was described and inappropriate (patients were allocated based on where they lived or when they were born).

7. The method of double blinding was described and inappropriate (e.g., identical placebo not used; study personnel and/or study participant could identify the intervention).

*Systematic reviews* were divided into those investigating the association of the various NDDs with ID, and reviews investigating the effects of iron supplementation.

## Results

### Overview of studies (Fig 1)

Among the 54 studies selected for this review, 40 investigated ADHD, including 4 systematic reviews, 30 association studies and 6 treatment studies (2 RCTs, 3 case reports, 1 case series). 12 studies investigated ASD including 1 systematic review, 10 association studies (including 4 prevalence studies), and 1 treatment study (RCT). Two association but no treatment studies were found for FASD (Fig 1).

### Overview of association studies

Among the 30 association studies identified for ADHD, 22 showed an association with ID and 8 did not. Among the 10 association studies identified for ASD, 3 showed an association with ID and 3 did not. 4 studies showed prevalence rates of ID within groups of patients with ASD without further statistical evaluation of associations / correlations. For FASD, both association studies identified an association of ID with growth, an FASD related manifestation. Numbers and types of studies, cases investigated as well as sex distribution are shown in Table 2.

### Association ADHD and ID (Fig 2, S3 Table in S1 File and S4 Tables in S2 File)

Overall, 30 association studies were identified. All but 1 study [25] were carried out in children/adolescents. The age of children/adolescents enrolled in the various studies was 4–19 years. The main instruments employed for the diagnosis and severity grading of ADHD were the Conners Parent Rating Scale (CPRS), used in 18/30 studies, the Conners Teacher Rating Scale (CTRS) used in 10/30 studies, and the Kiddie Schedule for Affective Disorders and Schizophrenia (KSADS) used in 10/30 studies. The instruments used in the individual studies are indicated in S3 and S4 Tables in S2 File.

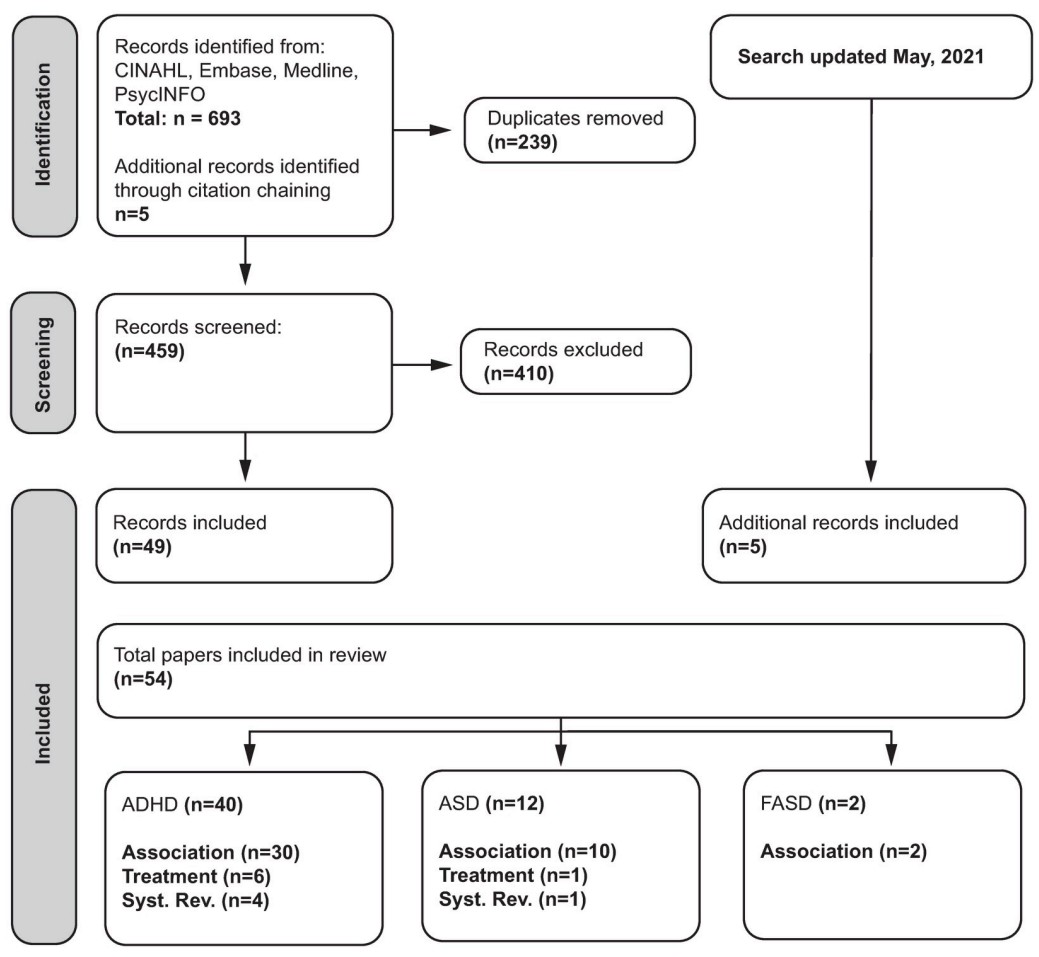

**Fig 1. Scoping review flow chart.**

An association of ADHD and/or ADHD severity with ID was shown in 22/30 studies including 1 cohort study, 12 case control studies and 8 case series, and 1 cross-sectional. Overall, 1018 cases were enrolled in the only cohort study, 1323 cases and 1124 controls were enrolled in case control studies, 1568 cases were enrolled in the case series, and 205 records analyzed in the cross-sectional study. 16/22 studies investigated the degree of ID and the severity of ADHD symptoms, with greater severity being associated with lower iron levels [11, 14, 25–38], and 10/22 studies concluded that SF levels were lower in the ADHD population [12, 13, 25, 30–32, 36, 39–41]. 10/22 studies showed an association with the hyperactivity symptoms of ADHD [25, 27–29, 32, 33, 35, 37, 38, 42]. Oppositional defiant/conduct disorder (10/22 studies) was the most common comorbidity in the ADHD populations investigated [14, 26–29, 31, 33, 34, 39, 42].

No association of ADHD and/or ADHD severity with ID was found in 8/30 studies (4 case control, 3 case series, and 1 cohort). 2805 cases were enrolled in the only cohort study showing no association between ADHD and ID [43], 251 cases and 203 controls were enrolled in the case control studies, and 458 cases were enrolled in case series studies. 7 studies investigated the degree of ID and severity of ADHD symptoms, with greater severity not being associated with lower iron levels [43–49], and 4 studies concluded that SF levels were not lower in the ADHD population [44, 46, 49, 50]. 6 studies showed lack of association with the hyperactivity

**Table 2. Overview of studies investigating the association of ID and an NDD and / or prevalence studies.**

| | Association | | | | No Association | | | Prevalence |
|---|---|---|---|---|---|---|---|---|
| | Case control | Case Series | Cohort | Cross Section | Case control | Case Series | Cohort | Cross Section |
| **ADHD** | | | | | | | | |
| Studies | 12 | 8 | 1 | 1 | 4 | 3 | 1 | |
| Cases | 1323 | 1568 | 1018* | 205 | 251 | 458 | 2805* | |
| Controls | 1124 | - - - - | - - - - | - - - - | 203 | - - - - | - - - - | |
| Cases m/f | 746/576 | 1279/219 | 519/499 | 137/68 | 212/39 | 355/103 | 1429/1376 | |
| **ASD** | | | | | | | | |
| Studies | 2 | 1 | | | 1 | 2 | | 4 |
| Cases | 408 | 96 | | | 154 | 64 | | 556 |
| Controls | 408 | - - - - | | | 73 | - - - - | | - - - - |
| Cases m/f | 237/171 | 78/18 | | | 141/13 | 54/10 | | 462/93 |
| **FASD** | | | | | | | | |
| Studies | 2 | | | | | | | |
| Cases | 127 | | | | | | | |
| Controls | - - - - | | | | | | | |
| Cases m/f | 61/66 | | | | | | | |

m/f = number of male/female patients.

Association = studies showing an association between NDD and ID; No Association = studies not showing an association between NDD and ID.

*total number of participants in cohort study.

Prevalence studies only reported about rates of ID in children with an NDD without further correlation studies.

symptoms of ADHD [43–46, 48, 49]. Oppositional defiant/conduct disorder (2/8 studies) was the most common comorbidity in the ADHD population investigated [44, 49].

## Association ASD and ID (Fig 3, S4–S6 Tables in S2 File)

Overall, 6 studies investigated associations between ID and ASD and 4 additional case series investigated the prevalence rates of ID in children with ASD. All studies were done in children

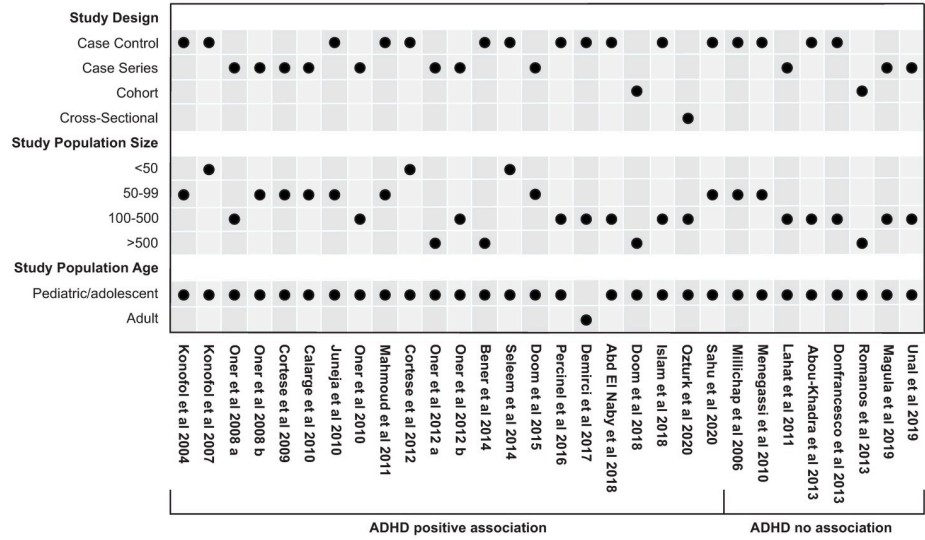

**Fig 2. Overview of ADHD association studies.** Pediatric/adolescent studies: Age 19 years or less; adult studies: Age 19 years or greater.

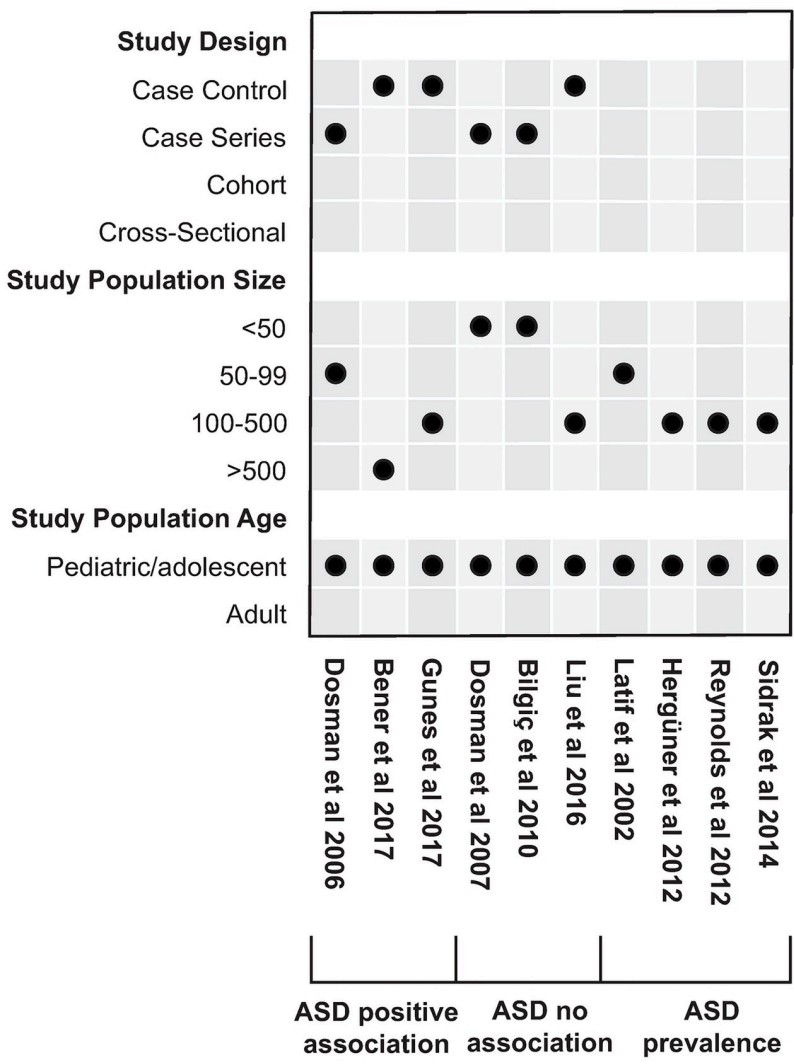

**Fig 3. Overview of ASD association studies.** Pediatric/adolescent studies: Age 19 years or less; adult studies: Age 19 years or greater.

and adolescents aged 1–18 years. Instruments employed for ASD assessment included the Autism Diagnostic Observation Schedule (ADOS) (4/10 studies), the Childhood Autism Rating Scale (CARS) (4/10 studies), Autism Behaviour Checklist (AuBC) (2/10 studies), and the Aberrant Behaviour Checklist (AbBC) (2/10 studies). The instruments used in the individual studies are indicated in S4–S6 Tables in S2 File.

3 studies (2 case control with 408 cases and 408 controls, and 1 case series with 96 cases enrolled) showed an association of ID and ASD and/or ASD-related symptoms. 3 studies (1 case control with 154 cases, 2 case series with 64 cases enrolled) did not. The 2 studies showing correlations between severity of ASD (as determined by the various scores used) and ID [17, 51] were contrasted by 3 other studies not showing such correlations [18, 52, 53]. The study by Dosman et al. [17] deserves a special explanation. The primary aim of this study was to investigate the effect of iron supplementation in children with ASD on sleep in an n-of-1 trial setting (each individual as their own control). As part of patient characterization at baseline, this study also analyzed the association between the degree of ID and ASD diagnostic scores.

Because no association was found between SF levels and total ADOS or ADI-R scores we included these results in the no association category. Nonetheless, it must be mentioned that in a subgroup of school-aged children, there was an inverse relationship between SF and the ADOS sub domain communication (p = 0.009). We did not include this study as a treatment study, as none of the ASD-specific scores was used as an outcome measure.

Lastly, among the 4 studies reporting ID prevalence in children diagnosed with ASD (enrolling 556 participants), low SF levels were found in 6.6–24% [16, 54–56].

## Association FASD and ID

We found 2 case control studies, including 85 and 42 cases, and 63 and 54 controls respectively, investigating the effect of heavy prenatal alcohol exposure on postnatal growth and postnatal ID compared to moderate/light exposure [57, 58]. Both studies found that in infants with heavy prenatal alcohol exposure, the extent of growth restriction was associated with ID, whereas there was no further modification of growth if ID or food insecurity persisted to the age of 5 years [58]. One of these studies [57] also showed an association of maternal binge drinking with an increased incidence of ID anemia at the age of 12 months. This study formally did not meet our inclusion criteria as only RBC-related markers were used to determine ID. We still included this study because of the paucity of studies identified for FASD in general and its close relation to the other study identified [58].

## Sex differences

Overall, there was a strong preponderance of male participants both in the ADHD and ASD studies (Table 2). The 30 ADHD association studies included 4677 males versus 2880 females (cases only). 12/30 studies examined potential sex differences in risk and severity of ADHD, while 6/30 studies examined potential sex differences in iron levels. 4 of these studies found that there were no significant differences between sex and ADHD symptoms [33, 39, 42, 47]. 5 studies found that males had more ADHD symptoms than females [11, 28, 34, 41, 50]. One study [26] compared iron status (iron sufficient, ID without anemia, ID anemia) and sex, and found a statistically significant difference (p<0.001) between the aforementioned groups. Similarly, Cortese et al. [14] reported lower serum ferritin levels in males compared to females, though this finding was not supported by any statistical analysis. The remaining 4 studies that examined sex differences and ID did not find any significant correlations [28, 39, 47, 50].

The 10 ASD association and prevalence studies included 972 males and 305 females (cases only). 2/10 studies examined potential sex differences in individuals with ASD. 1 study reported no significant differences between sex and ASD symptoms [16], while another study found a significant difference but did not elaborate on what this entailed [52]. 1 study analyzed differences between sex and iron status (including ferritin, MCV, hematocrit (Hct), hemoglobin), but did not find any statistically significant difference between males and females [16].

## Iron markers used in ADHD and ASD association studies (Figs 4 and 5, S1 Table, S2, S3 Tables in S1 File, S4-S6 Tables in S2 File)

Apart from 4 ADHD studies which measured only SF as a primary iron marker [41–43, 46] and 1 ADHD study which measured only SF and SI [50], all studies measured both iron-related and RBC-related markers for ID. A significant association of SF, but not of RBC-related or other iron markers was found in 9 ADHD [14, 28–32, 35, 37, 39] and 2 ASD studies [17, 54].

Overall, the most common indicators of statistically significant positive associations in studies were SF (n = 21) [12–14, 17, 25, 27–34, 36, 37, 39–42, 48, 59], followed by Hb (n = 8)

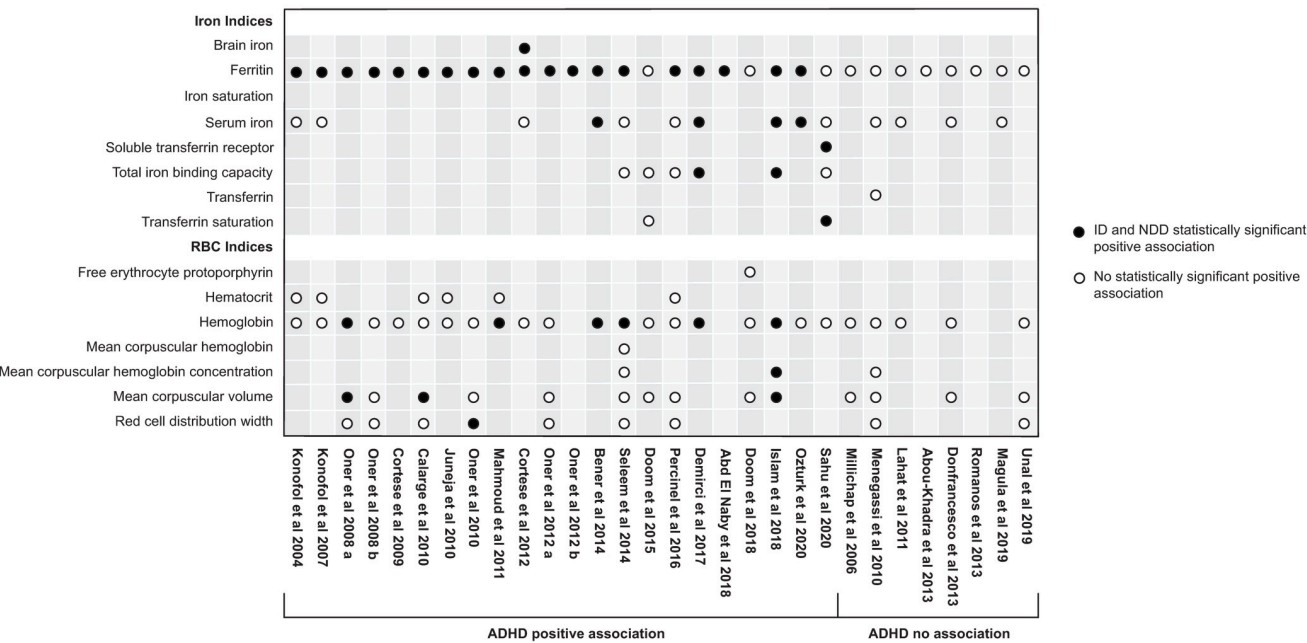

**Fig 4. Overview of biomarkers used in ADHD association studies.**

[12, 13, 17, 30, 33, 40, 51, 59], SI (n = 6) [12, 25, 37, 40, 51, 59], MCV (n = 5) [27, 33, 40, 51, 59], Hct (n = 2) [51, 59], and total iron binding capacity (TIBC) (n = 2) [25, 40]. Brain iron (BI) [39], mean corpuscular hemoglobina concentration (MCHC) [40], red cell distribution width (RDW) [34], sTfR/log ferritin index (SFI) [38], soluble transferrin receptor (sTfR) [38],

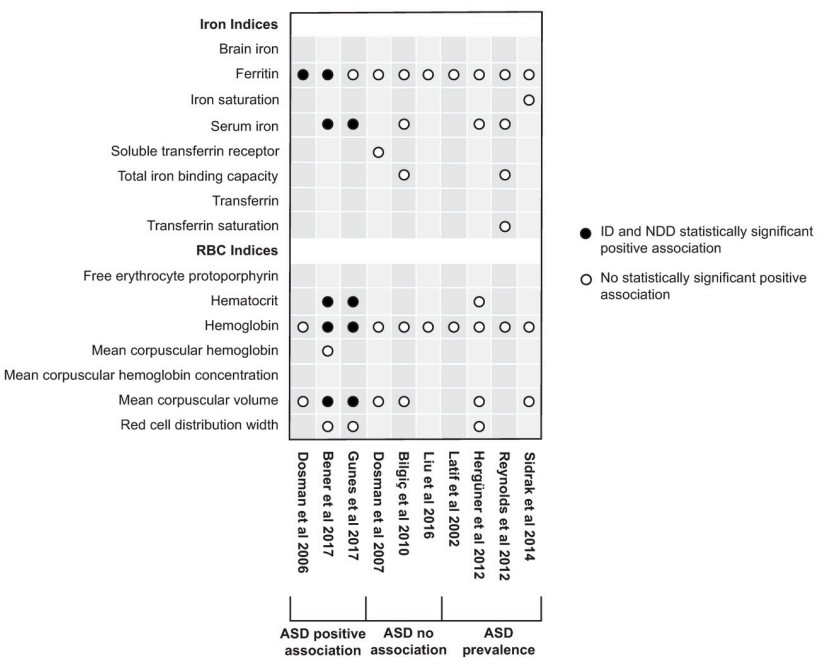

**Fig 5. Overview of biomarkers used in ASD association studies.**

and transferrin saturation (TSAT) [38], were indicators of a statistically significant positive association in 1 study each. The remaining studies which used the aforementioned biomarkers, but which did not represent a statistically significant positive association with ADHD or ASD are presented in Figs 4 and 5. Additional biomarkers used in the association studies include free erythrocyte protoporphyrin (FEP), transferrin (TF), and mean corpuscular hemoglobin (MCH).

In the 4 prevalence studies, the most commonly employed ID biomarkers were SF (n = 4), SI (n = 2), TSAT (n = 1), TIBC (n = 1), iron saturation (n = 1), Hb (n = 4), MCV (n = 2), Hct (n = 1), and RDW (n = 1) (Fig 5).

4 association studies [38, 43, 52, 59] measured c reactive protein (CRP) in conjunction with SF levels, while only 1 association study measured erythrocyte sedimentation rate (ESR) in addition to CRP [38]. The remainder of studies did not state the use of such inflammatory markers.

## Serum ferritin (SF) cut off-values (S3 Table in S1 File, S4-S6 Tables in S2 File, S7, S8 Tables in S3 File)

Among the 30 ADHD studies, SF cut-off values were reported in 20 studies. 7 studies used <12 µg/L and 13 studies used a range between <7 and 45 µg/L as cut-off value depending on the age of participants and / or presence or absence of high CRP levels. Among the 10 ASD studies, SF cut off levels were reported in 9, with cut off at < 10 µg/L for preschoolers and <12 µg/L for older children. Studies which did not report SF cut-off values, investigated correlations between SF and severity of ADHD / ASD scores without defining a ferritin-based threshold for ID.

## Fasting state for iron markers

Only 11 association studies identified that fasting blood samples were taken for measuring iron parameters [16, 28, 29, 31, 33, 35, 36, 40, 42, 45, 51]. In the remainder, no information on timing of blood sampling was given.

## Treatment studies (S7, S8 Tables in S3 File)

Studies investigating the effect of iron supplementation were available for ADHD and ASD. No such intervention studies were found for FASD. Overall, for ADHD 6 studies (2 RCTs and 3 case reports, and 1 case series) investigated the effects of iron supplementation. All of them demonstrated a benefit in treating symptoms with supplemental iron [15, 60–64], which includes 2 RCTs [15, 60].

The first RCT [15] included 22 (17 male, 5 female) non-anemic children (17 treated; 5 placebo) who had SF levels <30 ng/mL. While there was no significant improvement in the primary outcome measure (CPRS) after 12 weeks of oral ferrous sulfate supplementation (80 mg/ d) in the treatment group, there was a significant improvement of the ADHD Rating Scale (ADHD-RS), which was chosen as a secondary outcome measure. Both the hyperactivity and the inattention subscales improved, but the improvement was more pronounced in the inattention subscale. A similar result was observed in the second RCT, which included 42 children (35 male, 7 female) treated with methylphenidate and serum ferritin levels <30 ng/mL [60]. 21 children (16 male, 5 female) were treated with additional ferrous sulphate, 21 served as controls (19 male, 2 female). The study showed that the addition of ferrous sulfate resulted in an increase of SF levels and improvements in both hyperactivity and inattention symptomatology, measured using Child Symptoms Inventory-4 (CSI-4).

For ASD, 1 RCT was found, which included 20 children (17 male, 3 female) with insomnia. 9 children (6 male, 3 female) were treated with ferrous sulfate, 11 served as controls [65]. No significant differences were found in behaviour, as was measured using the SNAP-IV scale.

## Systematic reviews

We identified 5 systematic reviews, of which 4 investigated the association of iron indices (SF) with ADHD [66–69] and 1 with ASD [70]. The numbers and types of studies reviewed, and the conclusions are shown in Table 3. Overall, the 3 systematic reviews on ADHD and ID concluded that there is an association with SF levels (lower levels being associated with ADHD), while one systematic review concluded that the association between ADHD pathophysiology and ID has more to do with brain, rather than systemic iron [69]. The ASD study found no

**Table 3. Systematic literature reviews analyzing the role of iron deficiency in ADHD and ASD.**

| Reference | Review | Conclusion | Comments |
|---|---|---|---|
| **Systematic reviews** | | | |
| **N = 5** | | | |
| Scassellati et al | ADHD | *Descriptive* | Not significant after Bonferroni correction |
| 2012 [66] | Systematic Review | SF is a potential biomarker for ADHD | |
| | N = 7 | | |
| | case control studies | *Meta-analysis* | |
| | | SF is lower in patients with ADHD | |
| Wang et al | ADHD | *Descriptive* | |
| 2017 [67] | Systematic Review | SF is lower in patients with ADHD | |
| | N = 11 | | |
| | case control studies | | |
| Cortese et al | ADHD | *Descriptive* | No correlation with SI |
| 2012 [68] | Systematic Review | 22 studies reviewed used SF as biomarker for ID; | |
| | N = 22 | | |
| | Cohort, case control, cross-sectional, clinical trials, RCTs | Significant correlation between low SF and ADHD in 9/22 studies using SF as biomarker | |
| Tseng et al | ASD | *Descriptive* | |
| 2018 [70] | Scoping Review | No association of iron indices (including SF) with ASD | |
| | N = 25 | | |
| | Cohort and cross sectional studies | | |
| | Multiple iron indices (SF< SI< TSAT, TIBC, hair iron, iron in food) | *Meta-analysis* | |
| | | Levels of SF in children with ASD did not differ significantly from children without ASD | |
| Degremont et al | ADHD | *Descriptive* | |
| 2021 [69] | Systematic Review | 18 studies measured SF; 10/18 found higher SF in those with ADHD compared to controls | |
| | N = 20 | | |
| | Case control | 10 studies measured SI; 2/10 reported lower SI in the ADHD group | |
| | | 3 studies measured brain iron; all reported lower brain iron in the ADHD group | |
| Our review | ADHD, ASD, FASD | *Descriptive* | Heterogeneity of study designs, outcome measures and study participants. |
| | | Association with ID | |
| | Scoping Review | ADHD 22/30 | |
| | Association and treatment studies | ASD 3/6 | |
| | | FASD 2/2 | |
| | | Positive effect of iron supplementation | |
| | | ADHD 5/5 (2RCTs, 5 case reports) | |
| | | No effect of iron supplementation | |
| | | ASD 1/1 (RCT) | |

difference compared to controls. We did not identify any systematic literature review investigating the efficacy of iron supplementation.

## Year and location of studies

Association studies: The majority of the association studies (n = 19) were conducted in Europe [14, 16, 25, 28, 29, 32–37, 39, 42–44, 48, 51, 53, 54], followed by 7 in Asia [12, 31, 38, 40, 47, 52, 59], 6 in North America [11, 17, 18, 27, 49, 56], 5 in Africa [13, 30, 41, 46, 50], 2 in South America [26, 45], and 1 in Australia [55]. Moreover, these studies were predominantly (29/40) conducted in the 2010 decade [11–13, 16, 25–28, 30, 31, 34, 35, 39–48, 50–53, 55, 56, 59] with the remaining conducted in the 2020 [37, 38] and 2000 decades [14, 17, 18, 29, 32, 33, 36, 49, 54].

Treatment studies: The majority of the treatment studies were conducted in North America [61, 64, 65], followed by 2 in Europe [15, 62], and 2 in Asia [60, 63]. Moreover, 1 study was conducted in the 2020 decade [65], 2 conducted in the 2010 decade [60, 64], 2 conducted in the 2000 decade [15, 62], and 1 conducted in the 1990 decade [61].

## Discussion

We conducted this scoping review to obtain an overview of the role of ID in ADHD, ASD, and FASD. We chose these conditions not only because they represent the most frequently diagnosed NDDs in children, but also because they have the tendency to occur comorbidly (e.g. children with ASD often have phenotypical characteristics of ADHD and children with FASD often have phenotypical characteristics of ADHD and ASD). Furthermore, these conditions occur in combination with intellectual disability, which is prevalent in up to 3% of the population worldwide. While causal treatment is available only for a minority of intellectual disabilities [71], symptomatic treatment aiming to improve behaviours is a prerequisite to optimize residual developmental outcomes. The common clinical characteristics of the three NDDs investigated here are hypermotor restlessness and hyperarousability not only during the wake, but also during the sleep state, and hypoarousability during performance of cognitive tasks characterizing the circumscribed behavioural phenotypes (H-behaviours) [20, 72]. Together, these clinical characteristics suggest a common pathophysiology of the dopaminergic system for which iron is an important metabolic cofactor [73]. Our goal was to map the existing evidence as a basis for the development of guidelines for screening and treatment of ID in these conditions, and to develop a pragmatic best practice approach on how to further investigate these behavioural phenotypes. Our scoping review also helped to identify gaps in evidence generation and ideas to further analyse existing and to create prospective data.

In our review, ADHD has been found to be the most extensively investigated condition, whereas fewer studies have been conducted for ASD, and only 2 studies have been identified for FASD.

For ADHD, 22 of the 30 identified association studies suggested a relation of types and severity of ADHD-related symptoms with ID. These findings are supported by 4 systematic reviews concluding that ID plays a role in the pathogenesis of ADHD. We also found 8 studies which did not show evidence for an association of ID with ADHD.

There are several reasons for discrepant results in these ADHD association studies. One of them is the inconsistency in the use of instruments employed for the diagnosis and severity grading of ADHD. Overall, we identified 15 scales including a variety of self-report scales (which potentially capture internalizing behaviours that may go unnoticed by caregivers) and scales completed by adult informants (which mainly capture externalizing behaviours that are publicly observable). There are also differences in the bandwidth of the scales. Narrow band scales such as the CPRS and CTRS are robust measures to diagnose ADHD. In contrast,

broadband scales cover the breadth of patients' problems by eliciting information across an array of symptoms which can be associated with ADHD (e.g., Child Behaviour Checklist). For an overview see Collett et al [74].

When using narrow band scales, more than one baseline assessment should be performed to account for a change in scores related to treatment. However, in most of the ADHD association studies identified in our review, detailed information about the establishment of baseline values is not provided. Another limitation of the currently available scales is that they originally have been developed for male elementary school-aged children. Suitability to other populations investigated in the ADHD association studies reported here, e.g., preschoolers [75], girls [76], and children with comorbid intellectual disability and/or speech problems [77] has not been evaluated.

Inconsistencies in the choice of ID markers and interpreting ID results are additional concerns related to the discrepant results obtained in the ADHD association studies. In approximately three quarters of those studies which measured both iron- and RBC-related biomarkers, iron-related (predominantly SF), but not RBC-related markers were statistically associated with the diagnosis of ADHD and / or ADHD severity. For example, the case-control study by Konofal et al. [32], including 53 cases, and by Juneja et al. [31] including 25 cases, concluded a positive association between ID and ADHD. However, the positive association was only statistically significant for low SF levels in children with ADHD compared to controls, while SI, Hct, and Hb were in the normal range for both groups. Similar results were found in the series of 52 cases reported by Oner et al. [29], demonstrating a significant relationship between lower SF levels and ADHD symptom severity. Moreover, a case series by Cortese et al. [14] only conducted a statistical analysis on low and high SF level comparisons in students with ADHD, even though they had also used SI and Hb for the assessment of ID.

Similar to the ADHD association studies, contradictory results were also found in the ASD studies. Although the 6 association studies were comparable in study design and participant numbers, 3 found statistical evidence for an association of ID and ASD / ASD severity, and 3 did not. Further contradictory results were reported in the 4 studies investigating the prevalence of ID in children with ASD. While Hergüner et al. [16] found a prevalence of ID and IDA in 24.1% and 15.5% of the 116 ASD children investigated, a similar sized study by Sidrak et al. [55] found ID and IDA in only 6.6% and 4.1% respectively. Selective eating habits and gastrointestinal comorbidities, which are frequently observed in ASD may be one reason for these discrepancies [78, 79] along with discrepancies in the choice of tools and indicators to assess the severity of ASD and ID.

For FASD, we found only 2 studies with both of them investigating the association of ID and postnatal growth restriction in infants with heavy prenatal alcohol exposure [57, 58]. Although there is evidence from experimental studies showing that maternal iron status has a unique influence upon FASD outcomes [19, 80, 81], the role of maternal ID as a synergistic modifier of FASD risk has not been further investigated in humans. Studies investigating a potential correlation of maternal and neonatal ID with FASD-related neurodevelopmental outcomes as well as studies aiming to treat maternal ID are needed to demonstrate a potential FASD risk-reducing effect of intrauterine alcohol exposure in humans.

SF is currently recommended as the most practical, universally available biomarker for the detection of low iron stores [82]. One major advantage of SF over other primary iron markers is its widely accepted independence of the fasting / postprandial state, compared to SI levels and TSAT which are subject to postprandial elevations [83]. Indeed, in all studies analysed here, SF was used as a primary iron marker. Low SF levels were found to be most frequently associated with the NDDs investigated, whereas among the RBC-related iron markers, besides low Hb, low MCV was most frequently observed in association with NDDs. However, like

other RBC-related markers, MCV often remained unremarkable despite low SF levels. Likewise, most studies in our recently published scoping review about the role of ID in sleep disorders used SF, but not RBC-related measures as the main biomarker of ID [84].

Ferritin is a multimeric protein composed of light (L-ferritin) and heavy (H-ferritin) subunits which have distinct functions in oxidizing and binding (H-ferritin) and storing (L-ferritin) iron. While SF mainly contains L-ferritin for iron storage, the various ferritin species localized in cellular compartments, such as cytosol, golgi, lysosomes, and mitochondria, play roles in ferroptosis, a form of programmed cell death, ferrophagy, a form of autophagy, and handling of oxygen radicals [85].

Despite all advantages, the following limitations of SF's role as an indicator of ID should be considered in the interpretation of the results of this review. SF also acts as an acute phase protein and any increase in SF due to systemic inflammation confounds its relationship with ID. The parallel measurement of CRP as an independent indicator for acute inflammation enhances SF's value as an ID marker. In this review, CRP measurements in conjunction with serum ferritin were provided in only 2 ADHD [38, 43] and 2 ASD association studies [52, 59], while one study also measured erythrocyte sedimentation rate (ESR). In all 4 studies, none of the participants had an elevated CRP. However, the single study that measured both CRP and ESR found that ESR, but not CRP, was significantly higher in the ADHD group compared to controls (p = 0.001) [38]. The parallel measurement of CRP or ESR and / or the use of other biomarkers such as sTfR or sTfR/log ferritin ratio may be superior to SF alone, particularly in the presence of systemic inflammation [86]. Serum hepcidin is another promising biomarker indicating systemic iron regulation and inflammatory response, which could particularly useful in cases with ID resistant to enteral iron supplementation [87].

Another limitation is that SF represents iron stores mainly from macrophages, but it does not reflect ID contained to single organs [44]. For example, low brain iron (BI) levels, which are not always reflected by SF, play a role in the pathophysiology of restless legs syndrome (RLS), periodic limb movements in sleep [88], which can cause restless sleep disorders [89]. Given the presumptive overlap between RLS and ADHD, the use of BI imaging [39] has been recommended for a presumed subgroup of ADHD patients who may have deficient BI transport systems in the presence of normal SF levels. The finding of isolated BI deficiency in ADHD helps consolidating the concept of functional ID, as described in RLS [88, 90].

A final aspect in the interpretation of the results obtained in this review is the variance in normal SF values depending on age, sex and genotype. SF < 15 µg/L is generally accepted as the threshold for absence of iron stores [91], however, studies have shown that thresholds for ID in younger children are as low as <10–12 µg/L [92, 93]. Whereas the majority of the ASD studies analysed here used SF cut-off values were between 10 and 15 µg/L, SF cut-off values were higher than 15 µg/L in the majority of the ADHD studies.

The heterogeneous thresholds of SF as an indicator for ID might be another reason for discrepant results in the numerous studies performed among various age groups in countries across the globe. Also, if studies were performed in countries where infectious diseases are more common than in high resource countries (e.g., Africa, Asia South America), or during periods of seasonal infectious diseases, results based on SF only could have been confounded by its additional function as an acute phase protein.

Compared to the relative abundance of association studies, the yield for treatment studies in our search was low, with only 6 treatment studies for ADHD and 1 for ASD being identified. Although the 2 ADHD RCTs [15, 60] demonstrated a benefit of iron supplementation on both inattention and hyperactivity, they included low numbers of participants (18 treated cases vs 5 controls [15] and 21 treated cases vs 21 control [60]) and their Jaded scores were low at 2/ 5 and 3/5 respectively.

The overall paucity of ID treatment studies for ADHD and ASD stands in stark contrast to the abundance of clinical trials published for pharmacological treatments for ADHD. A scoping review investigating interventional ADHD RCTs that used sleep as an outcome measure identified 53 pharmacological RCTs published between 1995–2020 [94]. Importantly, this represents only a small proportion of pharmacological RCTs for ADHD, given that most studies do not use sleep as an outcome measure.

In the light of frequent adverse reactions [95] and of unknown long-term outcomes in stimulant users, well-designed and sufficiently powered clinical trials are needed to identify whether, and for which sub-populations iron supplementation can be a beneficial alternative or add on to pharmacotherapy for ADHD and other NDDs.

In conclusion, our scoping review identified a body of studies investigating a potential role of ID in frequent NDDs. However, the methodological designs, choice of outcome measures, and study participant numbers were quite heterogeneous. The paucity of treatment studies and the low numbers of enrolled participants is of particular concern given the high prevalence of the investigated NDDs and the easy access to iron supplementation.

While this review format allowed for a descriptive assessment of the identified studies, comparison of studies against each other was not attempted mainly due to the inconsistent methodological quality of the single studies. We accepted associations and related statistical results as described by the authors of the respective studies and did not exclude any study based on methodological design, participant numbers, choice of outcome measures for NDDs, or laboratory methods employed for measurement of iron markers.

The majority of studies identified in our scoping review supported the role of ID in NDDs, however there were also studies using similar study protocols, which did not. Furthermore, the paucity of treatment studies and the low numbers of enrolled participants deserves particular consideration, given the high prevalence of the investigated NDDs and the easy access to iron supplementation.

Given the global distribution of ID [22] paralleled by a similar distribution of NDDs [2], affecting both low and high resource countries, studies to corroborate the importance of prevention and treatment of ID are of utmost public health importance. In future studies, standardized protocols using harmonized outcome measures for symptom severity scores and iron indices will contribute to higher quality and consistency amongst studies. For treatment studies, besides adherence to high evidence study designs, the use N-of-1 trial designs with the treated individuals being their own control should be considered in single case studies, small sample sizes, and populations with heterogeneous clinical characteristics [96].

SF as the most accepted outcome measure for ID should be combined with iron markers independent of inflammation and / or with markers indicating inflammation and cut off levels should be harmonized. Additional markers addressing ID as a proxy for micronutrient deficiency or familial predispositions, and iron's spatial distribution (e.g., brain iron deficiency) should be developed and utilized particularly for studies investigating NDDs.

## Supporting information

**S1 Table. Preferred Reporting Items for Systematic reviews and Meta-Analyses extension for Scoping Reviews (PRISMA-ScR) checklist.** Adapted from: Tricco AC, Lillie E, Zarin W, O'Brien KK, Colquhoun H, Levac D, et al. PRISMA Extension for Scoping Reviews (PRISMAScR): Checklist and Explanation. Ann Intern Med. 2018;169:467–473. doi: 10.7326/M18-0850.
(DOCX)

**S1 File. S2 and S3 Tables.** Studies investigating the association of ADHD and ID.
(DOCX)

**S2 File. S4, S5, S6 Tables.** Studies investigating the association of ASD and ID.
(DOCX)

**S3 File. S7 and S8 Tables.** Treatment studies investigating the effect of iron supplementation in ADHD and ASD.
(DOCX)

## Acknowledgments

We thank Andrea Ryce, Librarian, BCCH for helping us with the online database search; Dr. Evelyn Stewart for funding the Iron Conundrum Workshop with the Michael Smith Foundation for Health Research Award 2017 (BCCH Research & Department of Psychiatry).

## Dedication

The authors would like to dedicate this article to Professor Richard P. Allen († Dec. 9, 2020), clinician and sleep scientist at Johns Hopkins University in Baltimore, Maryland, one of the pioneer investigators of Restless Legs Syndrome. Dr. Allen's iron deficiency research motivated us to review literature from different perspectives and approach day- and nighttime behaviours in unity.

## Author Contributions

**Conceptualization:** Ishmeet Singh, Wayne Leung, Sylvia Stockler, Osman S. Ipsiroglu.

**Data curation:** Scout McWilliams, Ishmeet Singh, Wayne Leung.

**Formal analysis:** Scout McWilliams, Ishmeet Singh, Sylvia Stockler.

**Investigation:** Scout McWilliams.

**Methodology:** Ishmeet Singh, Wayne Leung, Sylvia Stockler, Osman S. Ipsiroglu.

**Supervision:** Sylvia Stockler, Osman S. Ipsiroglu.

**Visualization:** Scout McWilliams.

**Writing – original draft:** Ishmeet Singh, Sylvia Stockler.

**Writing – review & editing:** Scout McWilliams, Ishmeet Singh, Sylvia Stockler, Osman S. Ipsiroglu.

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
