## [Decision Letter · Decision Letter 0]

8 Feb 2022

PONE-D-21-23571Iron deficiency and common neurodevelopmental disorders - A scoping reviewPLOS ONE

Dear Dr. Stockler,

Thank you for submitting your manuscript to PLOS ONE. After careful consideration, we feel that it has merit but does not fully meet PLOS ONE’s publication criteria as it currently stands. Therefore, we invite you to submit a revised version of the manuscript that addresses the points raised during the review process.

Both reviewers agree that the manuscript has significant merit and are positive towards publication, but both present a series of comments and concerns aiming at improving the manuscript that should be thoroughly addressed before final acceptance.

We look forward to receiving your revised manuscript.

Kind regards,

Efthimios M. C. Skoulakis, PhD

Academic Editor

PLOS ONE

https://journals.plos.org/plosone/s/file?id=ba62/PLOSOne_formatting_sample_title_authors_affiliations.pdf".

“SM was supported by a summer studentship from BCCHR, IS was supported by funds from the BCCHRI for sleep wake behaviour research (to OI) and for treatable intellectual disability research (to SS). WL was supported by the Mach-Gaensslen Foundation. The Iron Conundrum Workshop was funded by an award from the Michael Smith Foundation for Health Research.”

“SM was supported by a summer studentship from BC Children's Hospital Research Institute (https://www.bcchr.ca/), IS was supported by funds from the BC Children's Hospital Research Institute for sleep wake behaviour research (to OI) and for treatable intellectual disability research (to SS). WL was supported by the Mach-Gaensslen Foundation (https://mach-gaensslen.ca/). The Iron Conundrum Workshop was funded by an award from the Michael Smith Foundation for Health Research (https://www.msfhr.org/).

Reviewers' comments:

Reviewer's Responses to Questions

**Comments to the Author**

1. Is the manuscript technically sound, and do the data support the conclusions?

Reviewer #1: Yes

Reviewer #2: Yes

2. Has the statistical analysis been performed appropriately and rigorously? 

Reviewer #1: N/A

Reviewer #2: N/A

3. Have the authors made all data underlying the findings in their manuscript fully available?

Reviewer #1: Yes

Reviewer #2: Yes

4. Is the manuscript presented in an intelligible fashion and written in standard English?

Reviewer #1: Yes

Reviewer #2: Yes

5. Review Comments to the Author

Reviewer #1: The manuscript presents a scoping review of existing evidence that low serum ferritin, likely due to iron deficiency, either during pregnancy or during child growth and development, correlates with ADHD severity of young patients.

Discussing the point of whether "iron deficiency" is a helpful term to describe under one umbrella the complex possibilities of altered iron metabolism in humans, where a diagnostic parameter suggests iron unavailability is certainly outside the scope of the present review. I offer this comment, that the medical community might benefit by considering many more markers of iron status and how they are interconnected in disease conditions instead of a simplistic trichotomy of iron deficiency/adequacy/overload, for future reflections by the authors, in memory of Professor Richard P. Allen to whom the manuscript is dedicated. The authors themselves suggest the importance of measuring hepcidin and I have a question below why they seem skeptical about serum iron. In terms of the relationship between low serum ferritin and iron deficiency, there are a few interesting papers, and also there is literature around the physiologic function of serum ferritin. Should these points be discussed, given the much stronger evidence of correlation for this marker? The decision should be left with the authors in my opinion.

Other minor comments:

1. Inclusion and Exclusion criteria

Why do the authors exclude measurements of serum iron as primarily reflecting iron metabolism?

2. Association studies:

Were any criteria considered/applied to accept/exclude reported correlations?

3. The sentence in lines 177-188 could benefit from rephrasing so as not to repeat the word “studies” thrice.

4. Line 197: should the authors add an explanatory note of what they consider inappropriate methodology for double blinding?

5. Lines 234-244: would this information be best included in Supplementary Tables 1 & 2 with reference to the table in the text? Alternatively, this could also be done in their figures. I would have personally benefited by a concise descriptive comparison of these methods in the introduction, but this is a suggestion open to the authors to consider. If there is an available review, it could be a single introductory sentence of the paragraph where the methods are first mentioned.

6. Lines 414-415: order trials in ascending or descending temporal order?

7. Abstract:

“Conclusion: Screening for iron deficiency and use of iron supplementation is useful for neurodevelopmental disorders. Evidence is higher for ADHD, lower for ASD, and needs to be further investigated for FASD.”

Suggested change in wording:

“Conclusion: Evidence in favor of screening for iron deficiency and using iron supplementation for pediatric neurodevelopmental disorders comes primarily from ADHD studies and needs to be further investigated for ASD and FASD.”

Reviewer #2: In this scoping, the authors present an investigation of the association of iron deficiency (ID) in children and adolescents with the most frequent neurodevelopmental disorders (NDD), ADHD, ASD, and FASD. They also assess whether iron supplementation improves outcomes in these disorders. This is set on the backdrop of a role of ID in NDDs, but a paucity of evidence for the impact of ID and NDDSl disorders in childhood and adolescence. Particularly, that postnatal ID may aggravate pre-existing NDDs. This is an interesting and, seemingly novel review.

Introduction:

Well written, providing good background and context to the study. One missing area relates to the global distribution of ID and NDDs. There is plentiful data mapping the global distribution of ID, showing greatest prevalence in low-resource settings of the world, but also among at risk sub-groups. Is there any similar parallel with the global distribution of NDDs?

Methods

Was the review pre-registered on PROSPERO?

I can’t see the age limitation of children for included vs. excluded studies. Where is this information?

Results

The results are clearly presented.

The sex differences reported (lines 314 onwards), how did these differences relate to iron status. As this information is presented separately, it is difficult to work this out. Boys are at risk of iron deficiency in contexts of food insecurity, especially during periods of rapid growth. Are boys at greater risk because of differential iron status, rather than other sex-specific factors? It would be good to see the results a little more linked up.

On a similar point, it was hard to find details on actual iron status of the populations under investigation. This could be key to interpreting the findings (e.g. no associations may be observed in populations where there is not sufficient variance in iron status or in populations where status does not cross a ‘threshold’ for deficiency to impact on the outcomes under investigation. I have looked at all the Tables in the supporting information and this detail is also missing from there. I think it could be useful and potentially informative to position the papers against each other with respect to iron status of the children included. This could be done as a Figure, potentially showing levels of iron deficiency anaemia in each context.

Discussion

The discussion is well written and provides a good commentary on the results presented.

However, I was left wondering how much of the evidence for any associations presented was (i) because of iron deficiency per se (e.g. not because of nutritional insufficiency, with iron status being a proxy marker for global micronutrient deficiency) and (ii) how much of the evidence was confounded by other familial risk factors. I can see the benefit of including all types of study design in a scoping review and where trial evidence is limited. However, I felt that insufficient weighting was given to the weight of the evidence being presented and, therefore, the associated limitations. This is linked in with the point above about teasing out causal pathways; I think there is scope within this review to do this more thoroughly.

6. PLOS authors have the option to publish the peer review history of their article (what does this mean?). If published, this will include your full peer review and any attached files.

Reviewer #1: **Yes: **Fanis Missirlis

Reviewer #2: No

---

## [Author Response · Author response to Decision Letter 0]

8 Aug 2022

Reviewer #1: The manuscript presents a scoping review of existing evidence that low serum ferritin, likely due to iron deficiency, either during pregnancy or during child growth and development, correlates with ADHD severity of young patients.

Discussing the point of whether "iron deficiency" is a helpful term to describe under one umbrella the complex possibilities of altered iron metabolism in humans, where a diagnostic parameter suggests iron unavailability is certainly outside the scope of the present review. I offer this comment, that the medical community might benefit by considering many more markers of iron status and how they are interconnected in disease conditions instead of a simplistic trichotomy of iron deficiency/adequacy/overload, for future reflections by the authors, in memory of Professor Richard P. Allen to whom the manuscript is dedicated. The authors themselves suggest the importance of measuring hepcidin and I have a question below why they seem skeptical about serum iron. In terms of the relationship between low serum ferritin and iron deficiency, there are a few interesting papers, and also there is literature around the physiologic function of serum ferritin. Should these points be discussed, given the much stronger evidence of correlation for this marker? The decision should be left with the authors in my opinion.

Answer: We addressed this important aspect in the last paragraph of the discussion (page 31): 

SF as the most accepted outcome measure for ID should be combined with iron markers independent of inflammation and / or with markers indicating inflammation and cut off levels should be harmonized. Additional markers addressing iron’s spatial distribution (e.g., brain iron determined via in vivo magnetic resonance technology) and ferritin’s various functions, should be developed and utilized particularly for studies investigating NDDs.

Other minor comments:

1. Inclusion and Exclusion criteria

Why do the authors exclude measurements of serum iron as primarily reflecting iron metabolism?

Answer: 

Our main rationale is that serum iron is susceptible to post-prandial elevation, making it an unreliable marker of iron status unless fasting bloods are taken. However, we would like to highlight that none of the studies identified in our search were excluded for this reason. We have mentioned serum iron’s dependence on the postprandial state in the discussion section on page 26: 

SF is currently recommended as the most practical, universally available biomarker for the detection of low iron stores [81]. One major advantage of SF over other primary iron markers is its widely accepted independence of the fasting / postprandial state, compared to SI levels and TSAT which are subject to postprandial elevations [80].

2. Association studies:

Were any criteria considered/applied to accept/exclude reported correlations?

Answer: we have addressed this question in the limitations section of the discussion (page 30).

While this review format allowed for a descriptive assessment of the identified studies, comparison of studies against each other was not attempted mainly due to the inconsistent methodological quality of the single studies. We accepted associations and related statistical results as described by the authors of the respective studies and did not exclude any study based on methodological design, participant numbers, choice of outcome measures for NDDs, or laboratory methods employed for measurement of iron markers. 

3. The sentence in lines 177-188 could benefit from rephrasing so as not to repeat the word “studies” thrice. 

Answer: This sentence has been rephrased. 

4. Line 197: should the authors add an explanatory note of what they consider inappropriate methodology for double blinding? 

Answer: We have added an explanatory note of what inappropriate double blinding could look like (page 10-11): 

7) The method of double blinding was described and inappropriate (e.g., identical placebo not used; study personnel and/or study participant could identify the intervention). 

5. Lines 234-244: would this information be best included in Supplementary Tables 1 & 2 with reference to the table in the text? Alternatively, this could also be done in their figures. I would have personally benefited by a concise descriptive comparison of these methods in the introduction, but this is a suggestion open to the authors to consider. If there is an available review, it could be a single introductory sentence of the paragraph where the methods are first mentioned.

Answer:

We have added the tools used to determine the severity of ADHD and ASD to Tables S2, S3 for ADHD and S4, S5, S6 for ASD) to make this section more concise and easier to follow. We also have created a paragraph in the discussion describing the characteristics and differences of the ADHD tools (page 24-25): 

Overall, we identified 15 scales including a variety of self-report scales (which potentially capture internalizing behaviours that may go unnoticed by caregivers) and scales completed by adult informants (which mainly capture externalizing behaviours that are publicly observable). There are also differences in the bandwidth of the scales. Narrow band scales such as the CPRS and CTRS are robust measures to diagnose ADHD. In contrast, broadband scales cover the breadth of patients’ problems by eliciting information across an array of symptoms which can be associated with ADHD (e.g., Child Behaviour Checklist). For an overview see Collett et al [75]. 

When using narrow band scales, more than one baseline assessment should be performed to account for a change in scores related to treatment. However, in most of the ADHD association studies identified in our review, detailed information about the establishment of baseline values is not provided. Another limitation of the currently available scales is that they originally have been developed for male elementary school-aged children. Suitability to other populations investigated in the ADHD association studies reported here, e.g., preschoolers [76], girls [77], and children with comorbid intellectual disability and/or speech problems [78] has not been evaluated . 

6. Lines 414-415: order trials in ascending or descending temporal order?

We have re-ordered these trials. 

7. Abstract:

“Conclusion: Screening for iron deficiency and use of iron supplementation is useful for neurodevelopmental disorders. Evidence is higher for ADHD, lower for ASD, and needs to be further investigated for FASD.”

Suggested change in wording:

“Conclusion: Evidence in favor of screening for iron deficiency and using iron supplementation for pediatric neurodevelopmental disorders comes primarily from ADHD studies and needs to be further investigated for ASD and FASD.”

Answer: Thank you for this suggestion - we have used this in the amended conclusion sentence. 

Reviewer #2: In this scoping, the authors present an investigation of the association of iron deficiency (ID) in children and adolescents with the most frequent neurodevelopmental disorders (NDD), ADHD, ASD, and FASD. They also assess whether iron supplementation improves outcomes in these disorders. This is set on the backdrop of a role of ID in NDDs, but a paucity of evidence for the impact of ID and NDDSl disorders in childhood and adolescence. Particularly, that postnatal ID may aggravate pre-existing NDDs. This is an interesting and, seemingly novel review.

Introduction:

Well written, providing good background and context to the study. One missing area relates to the global distribution of ID and NDDs. There is plentiful data mapping the global distribution of ID, showing greatest prevalence in low-resource settings of the world, but also among at risk sub-groups. Is there any similar parallel with the global distribution of NDDs?

Answer: Thank you for pointing this out. We have found a paper entitled Global, regional, and national burden of 12 mental disorders in 204 countries and territories, 1990–2019: a systematic analysis for the Global Burden of Disease Study 2019, which we have now referenced in the introduction (page 4) and the discussion (page 31): 

Given the global distribution of ID [22] paralleled by a similar distribution of NDDs [2], affecting both low and high resource countries, future studies to corroborate the importance of prevention and treatment of ID are of utmost public health importance.

Methods

Was the review pre-registered on PROSPERO?

Answer: PROSPERO does not accept scoping reviews, therefore we were not able to register it. We have, however, included the PRISMA-Scr Checklist (Table S1) to further structure our paper. 

I can’t see the age limitation of children for included vs. excluded studies. Where is this information?

Answer: We did not include any age limitations in our inclusion/exclusion criteria. However, we realized that several sentences in the manuscript were referring only to children (when in fact they should have been referring to both children and adults). We have revised this wording where necessary. 

Results

The results are clearly presented. The sex differences reported (lines 314 onwards), how did these differences relate to iron status. As this information is presented separately, it is difficult to work this out. Boys are at risk of iron deficiency in contexts of food insecurity, especially during periods of rapid growth. Are boys at greater risk because of differential iron status, rather than other sex-specific factors? It would be good to see the results a little more linked up.

Answer: Thank you for this comment. We have re-analyzed the studies and have updated our results to reflect those studies which looked at any correlation between ID and sex/gender (page 16-17): 

Overall, there was a strong preponderance of male participants both in the ADHD and ASD studies (Table 2). The 30 ADHD association studies included 4677 males versus 2880 females (cases only). 12/30 studies examined potential sex differences in risk and severity of ADHD, while 6/30 studies examined potential sex differences in iron levels. 4 of these studies found that there were no significant differences between sex and ADHD symptoms [33,39,42,47]. 5 studies found that males had more ADHD symptoms than females [11,28,34,41,50]. One study [26] compared iron status (iron sufficient, ID without anemia, ID anemia) and sex, and found a statistically significant difference (p<0.001) between the aforementioned groups. Similarly, Cortese et al. [14] reported lower serum ferritin levels in males compared to females, though this finding was not supported by any statistical analysis. The remaining 4 studies that examined sex differences and ID did not find any significant correlations [28,39,47,50]. 

The 10 ASD association and prevalence studies included 972 males and 305 females (cases only). 2/10 studies examined potential sex differences in individuals with ASD. 1 study reported no significant differences between sex and ASD symptoms [16], while another study found a significant difference but did not elaborate on what this entailed [52]. 1 study analyzed differences between sex and iron status (including ferritin, MCV, hematocrit (Hct), hemoglobin), but did not find any statistically significant difference between males and females [16]. 

On a similar point, it was hard to find details on actual iron status of the populations under investigation. This could be key to interpreting the findings (e.g. no associations may be observed in populations where there is not sufficient variance in iron status or in populations where status does not cross a ‘threshold’ for deficiency to impact on the outcomes under investigation. I have looked at all the Tables in the supporting information and this detail is also missing from there. I think it could be useful and potentially informative to position the papers against each other with respect to iron status of the children included. This could be done as a Figure, potentially showing levels of iron deficiency anaemia in each context.

Answer: We were not able to find information about the iron status of the populations investigated. However, we now included both the country in which the respective studies were performed and the serum ferritin cut off values chosen in the various studies in tables S2, S3 for ADHD association studies and in S4-S6 for ASD studies. We also appreciate the importance of the recommended cut off levels for SF concentrating as indicators for iron deficiency as well as population-specific confounders in the discussion (page 28-29):

A final aspect in the interpretation of the results obtained in this review is the variance in normal SF values depending on age, sex and genotype. SF < 15 �g/L is generally accepted as the threshold for absence of iron stores [92], however, studies have shown that thresholds for ID in younger children are as low as <10-12 �g/L [93,94]. Whereas the majority of the ASD studies analysed here used SF cut-off values were between 10 and 15 �g/L, SF cut-off values were higher than 15 �g/L in the majority of the ADHD studies. 

The heterogeneous thresholds of SF as an indicator for ID might be another reason for discrepant results in the numerous studies performed among various age groups in countries across the globe. Also, if studies were performed in countries where infectious diseases are more common than in high resource countries (e.g., Africa, Asia South America), or during periods of seasonal infectious diseases, results based on SF only could have been confounded by its additional function as an acute phase protein. 

Discussion

The discussion is well written and provides a good commentary on the results presented.

However, I was left wondering how much of the evidence for any associations presented was (i) because of iron deficiency per se (e.g. not because of nutritional insufficiency, with iron status being a proxy marker for global micronutrient deficiency) and (ii) how much of the evidence was confounded by other familial risk factors. I can see the benefit of including all types of study design in a scoping review and where trial evidence is limited. 

Answer: We addressed the possibility that ID is only a proxy marker for nutritional deficiencies and familial risk factors in the discussion (page 31): 

SF as the most accepted outcome measure for ID should be combined with iron markers independent of inflammation and / or with markers indicating inflammation and cut off levels should be harmonized. Additional markers addressing ID as a proxy for micronutrient deficiency or familial predispositions, and iron’s spatial distribution (e.g., brain iron deficiency) should be developed and utilized particularly for studies investigating NDDs.

However, I felt that insufficient weighting was given to the weight of the evidence being presented and, therefore, the associated limitations. This is linked in with the point above about teasing out causal pathways; I think there is scope within this review to do this more thoroughly.

Answer: We addressed the limitations of our review in weighing the evidence presented in the discussion (page 30):

While this review format allowed for a descriptive assessment of the identified studies, comparison of studies against each other was not attempted mainly due to the inconsistent methodological quality of the single studies. We accepted associations and related statistical results as described by the authors of the respective studies and did not exclude any study based on methodological design, participant numbers, choice of outcome measures for NDDs, or laboratory methods employed for measurement of iron markers.

---

## [Decision Letter · Decision Letter 1]

17 Aug 2022

Iron deficiency and common neurodevelopmental disorders - A scoping review

PONE-D-21-23571R1

Dear Dr. Stockler,

We’re pleased to inform you that your manuscript has been judged scientifically suitable for publication and will be formally accepted for publication once it meets all outstanding technical requirements.

Kind regards,

Efthimios M. C. Skoulakis, PhD

Academic Editor

PLOS ONE

Additional Editor Comments (optional):

Reviewers' comments:

Reviewer's Responses to Questions

**Comments to the Author**

1. If the authors have adequately addressed your comments raised in a previous round of review and you feel that this manuscript is now acceptable for publication, you may indicate that here to bypass the “Comments to the Author” section, enter your conflict of interest statement in the “Confidential to Editor” section, and submit your "Accept" recommendation.

Reviewer #1: All comments have been addressed

2. Is the manuscript technically sound, and do the data support the conclusions?

Reviewer #1: Yes

3. Has the statistical analysis been performed appropriately and rigorously? 

Reviewer #1: N/A

4. Have the authors made all data underlying the findings in their manuscript fully available?

Reviewer #1: Yes

5. Is the manuscript presented in an intelligible fashion and written in standard English?

Reviewer #1: Yes

6. Review Comments to the Author

Reviewer #1: The authors have responded to all comments in a considerate and careful manner. Their scoping review should be of interest and help to the community of researchers and medical professionals that work with attention deficit hyperactivity disorder, autism, and fetal alcohol syndrome. The role of iron metabolism in ADHD is well highlighted by the present review.

7. PLOS authors have the option to publish the peer review history of their article (what does this mean?). If published, this will include your full peer review and any attached files.

Reviewer #1: **Yes: **Fanis Missirlis

---

## [Editor Report · Acceptance letter]

19 Sep 2022

PONE-D-21-23571R1 

Iron deficiency and common neurodevelopmental disorders – A scoping review 

Dear Dr. Stockler:

I'm pleased to inform you that your manuscript has been deemed suitable for publication in PLOS ONE. Congratulations! Your manuscript is now with our production department. 

Kind regards, 

on behalf of

Dr. Efthimios M. C. Skoulakis 

Academic Editor

PLOS ONE